# Effects of *Hevea brasiliensis* Intercropping on the Volatiles of *Pandanus amaryllifolius* Leaves

**DOI:** 10.3390/foods12040888

**Published:** 2023-02-19

**Authors:** Ang Zhang, Zhiqing Lu, Huan Yu, Yaoyu Zhang, Xiaowei Qin, Xunzhi Ji, Shuzhen He, Ying Zong, Yiming Zhong, Lihua Li

**Affiliations:** 1Spice and Beverage Research Institute, Chinese Academy of Tropical Agricultural Sciences, Wanning 571533, China; 2Hainan Provincial Key Laboratory of Genetic Improvement and Quality Regulation for Tropical Spice and Beverage Crops, Spice and Beverage Research Institute, Chinese Academy of Tropical Agricultural Sciences, Wanning 571533, China; 3Key Laboratory of Genetic Resource Utilization of Spice and Beverage Crops of Ministry of Agriculture and Rural Affairs, Spice and Beverage Research Institute, Chinese Academy of Tropical Agricultural Sciences, Wanning 571533, China; 4College of Tropical Crops, Yunnan Agricultural University, Pu’er 665000, China

**Keywords:** aroma components, food resources, *Hevea brasiliensis* (Willd. ex A. Juss.) Muell. Arg., intercropping, *Pandanus amaryllifolius* Roxb., pyrrole

## Abstract

*Pandanus amaryllifolius* Roxb. is a special tropical spice crop resource with broad development prospects. It is widely cultivated under a *Hevea brasiliensis* (Willd. ex A. Juss.) Muell. Arg. canopy to improve the comprehensive benefits to *Hevea brasiliensis* plantations in Hainan Provence, China. However, the effects of intercropping with *Hevea brasiliensis* on the component number and relative contents of volatile substances in different categories in the *Pandanus amaryllifolius* leaves are still unknown. Therefore, a *Hevea brasiliensis* and *Pandanus amaryllifolius* intercropping experiment was set up to clarify the differences between several cultivated patterns on volatile substances in the *Pandanus amaryllifolius* leaves, and the key regulatory factors of volatile substances. The results showed that the soil pH was significantly decreased, while soil bulk density, alkali-hydrolyzable nitrogen and available phosphorus contents were significantly increased under the intercropping pattern. The component numbers of esters in volatile substances were increased by 6.20%, while the component numbers of ketones were decreased by 4.26% under the intercropping pattern. Compared with the *Pandanus amaryllifolius* monoculture, the relative contents of pyrroles, esters and furanones were significantly increased by 8.83%, 2.30% and 8.27%, respectively, while the relative contents of ketones, furans and hydrocarbons were decreased by 1.01%, 10.55% and 9.16% under the intercropping pattern, respectively. The relative contents of pyrroles, esters, furanones, ketones, furans and hydrocarbons were associated with changes in soil pH, soil available phosphorus content and air temperature. The results indicated that the reduction in soil pH and enhancement in soil-available phosphorus may be the main reasons for promoting the relative content of pyrroles and reducing the relative content of hydrocarbons under an intercropping pattern. Overall, *Hevea brasiliensis* intercropping with *Pandanus amaryllifolius* could not only improve soil properties, but also significantly increase the relative contents of the main volatile substances in *Pandanus amaryllifolius* leaves, which could provide a theoretical basis for the application and promotion of high-quality production patterns of *Pandanus amaryllifolius*.

## 1. Introduction

*Pandanus amarylifolius* Roxb. is a tropical perennial herb spice plant, which is native to the Maluku Islands of Indonesia, and widely distributed in South and Southeast Asian countries, such as India, Thailand and Malaysia. As the “Oriental Vanilla” with tropical characteristics, *Pandanus amaryllifolius* is widely used in the fields of food and medicine [1,2]. On the one hand, the leaves of *Pandanus amaryllifolius* are traditionally used as natural flavoring agents or food raw materials for candy, pastries, cakes and other foods, because they are an excellent source of 2-acetyl-1-pyrroline (2AP) [3,4]. Moreover, *Pandanus amaryllifolius* is also one of the natural green colorants because of the high content of chlorophylls (Chl a and Chl b) in its leaves [5]. On the other hand, an appreciable amount of active ingredients, such as phytol, squalene and linoleic acid are found in the leaves of *Pandanus amaryllifolius* [6,7,8,9], which become a good medicinal resource. The leaves of *Pandanus amaryllifolius* are used as drugs to treat diabetes in traditional medicine [10]. In addition, the mannose-binding protein in *Pandanus amaryllifolius* is conducive to enhance immune function [11]. It is noteworthy that the contents of volatiles, such as 2AP, squalene and phytol in *Pandanus amaryllifolius*, are significantly higher than those in other crops [12]. *Pandanus amaryllifolius* has become a hot resource in the field of biomedicine and natural product research because of its large quantity of cheap and easily available bioactive compounds. There is potential to increase the relative contents of active substances in *Pandanus amaryllifolius* by improving the planting pattern and technology, which may be a great benefit to promote the development of *Pandanus amaryllifolius* resources, and to expand the potential of *Pandanus amaryllifolius* high-quality production.

The original habitat of *Pandanus amaryllifolius* is lowland tropical rainforest, which has the characteristics of shade tolerance, and is suitable for plantation conditions [1,13]. As an important strategic material, *Hevea brasiliensis* (Willd. ex A. Juss.) Muell. Arg. has become one of the main economic crops in Southeast and South Asia [14]. However, the wide spacing of *Hevea brasiliensis* rows not only wastes land resources, but also limits the economic benefits of *Hevea brasiliensis* plantations. Intercropping is a cultivation pattern based on promoting and supplementing the plantation ecosystem [15], which can balance the spatial structure and interspecific competition of crops, optimize resource utilization [16], improve the spatio-temporal utilization of farmland soil [17], promote the healthy growth of crops and increase productivity [18], in order to obtain higher economic benefits from limited land resources [19]. Previous research indicated that intercropping with cash crops could significantly improve soil physico-chemical properties of *Hevea brasiliensis* plantations [20]. It can also markedly promote the photosynthesis of *Hevea brasiliensis* seedlings by 26.83% (Figure A1), based on the author’s previous observations. Therefore, a *Hevea brasiliensis* and *Pandanus amaryllifolius* intercropping pattern could promote the productivity of commercial crops and become an efficient production pattern of tropical plantations.

Previous studies indicated that using an intercropping pattern significantly regulated productivity and quality of crops by changing the soil physico-chemical properties, ecosystem micro-environment and regulating crop physiological processes [21]. For example, the biomass and production of wheat was improved after intercropping with broad beans [22]. The increase in soil nitrogen content improves the tea quality by accelerating the accumulation of amino acid and caffeine content in tea, under an intercropping pattern of soybean and tea [23]. After intercropping with soybean, mint productivity increased by 50%, and the quality of mint oil was improved through the increased relative content of menthol, and the decreased relative content of menthol furan and menthol acetate [24]. The main quality of *Pandanus amaryllifolius* is its volatile aroma components and active substances, such as pyrroles (2-Acetyl-1-Pyrroline), hydrocarbons (squalene, neophytadiene) and alcohols (Phytol). However, the volatile substances in *Pandanus amaryllifolius* are still unknown after intercropping with *Hevea brasiliensis*. In order to ascertain the effect of an intercropping pattern on the quality of Pa, this study addressed the following two scientific questions by comparing the differences in soil properties as well as volatile substances in *Pandanus amaryllifolius*, between monoculture and an intercropping pattern: 1. Explore the change rule of the component numbers and relative contents of volatile substances in different categories under a *Pandanus amaryllifolius* and *Hevea brasiliensis* intercropping pattern; 2. clarify the key regulatory factors affecting the relative contents of *Pandanus amaryllifolius* volatile substances in different categories under the intercropping pattern. The results could definite the main category and key regulatory factors of volatile substances in *Pandanus amaryllifolius* leaves under an intercropping pattern, which could provide a theoretical basis for the application and promotion of high-quality production of *Pandanus amaryllifolius*.

## 2. Materials and Methods

### 2.1. Materials

*Pandanus amaryllifolius* was planted in the Germplasm Resource Nursery of Spice and Beverage Research Institute, Chinese Academy of Tropical Agricultural Sciences (110°13′ E, 18°15′ N), Wanning City, Hainan Province, China. *Pandanus amaryllifolius* leaves were generally harvested after being transplanted to the field for 10 months. Thus, the *Pandanus amaryllifolius* leaves were collected after being transplanted and grown for 1 year in this study. The healthy leaves (usually the fifth to seventh fresh leaves from the top to the bottom) with the same growth, no pests and diseases were selected as experimental materials. Leaf samples were cut off at the base of the blade, and then transferred immediately to the laboratory for determination in October 2021.

### 2.2. Methods

#### 2.2.1. Sample Preparation

##### Growing Environment

The experimental region belonged to the tropical monsoon climate. The annual mean temperature, precipitation and sunshine hours were 22 °C, 2100–2200 mm and 1750–2650 h, respectively. The soil was tidal sand–mud (US Soil Taxonomy classification). The soil pH and organic matter content were 6.00 and 20.04 g·kg^−1^, respectively.

##### Experiment Design

Twenty 0.5 m × 1 m experimental plots were established with a randomized block design in this experiment. The blocks were arranged in ten rows with a 1 m buffer between each plot. Each block had two plots, and one plot was set for each planting pattern: *Pandanus amaryllifolius* monoculture (M), *Hevea brasiliensis* and *Pandanus amaryllifolius* intercropping (I). The plant spacing of different crops was 50 cm in each plot. Each treatment was replicated ten times. During the experiment, the management (e.g., fertilization, irrigation, shading, etc.) of all plots was consistent.

#### 2.2.2. Determination of Plant Samples

##### Sample Collection and Pretreatment

The *Pandanus amaryllifolius* leaves were immediately measured when they were returned to the laboratory. Firstly, the dirt and dust on the leaf surface were removed using gauze and distilled water. Secondly, the middle part of leaf was selected by manually shearing with scissors, in order to avoid the loss of volatile substances caused by the temperature increase during mechanical crushing. Thirdly, 5.0 g of broken fresh leaf were placed into a 50 mL centrifuge tube, and then ultrasonically extracted for 1 h under the conditions of 400 W, 50 °C and 40 KHz. After, 15 mL of absolute ethanol were added into the 50 mL centrifuge tube. Fourthly, the upper liquid was transferred to another 50 mL centrifuge tube via a 0.22 μm organic phase needle filter (nylon). Fifthly, 5 g of anhydrous sodium sulfate were added into the extracted solution samples to remove the water in the liquid. The supernatant was used to detect volatile substances of *Pandanus amaryllifolius*.

##### Qualitative Analysis

Qualitative analysis of volatile substances in the *Pandanus amaryllifolius* leaves was implemented by gas chromatography and mass spectrometry (GC–MS). The results of GC–MS analysis were retrieved from the NIST 2017 spectral library and compared for qualitative analysis: The Piano N-Paraffins Mix (C7~C40, Shanghai Amphora Experimental Technology Co., Ltd., Shanghai, China) was used to conduct GC–MS analysis with the same procedure, and calculate the retention index (RI) of each volatile component according to the linear equation using its retention time [25]. The actually measured RI was compared with that in the literature for qualitative analysis.

##### Quantitative Analysis

Accurately absorbed 2-acetyl-1-pyrroline (2AP) standard (purity 95%, Toronto Research Chemicals, Canada) and methanol (chromatographic purity) were configured according to 50, 100, 150, 300 and 500 μg mL^−1^ solution concentration gradients, separately. The standard gradient concentration was injected for GC–MS from low to high, according to the chromatographic conditions of the sample; each concentration was injected three times. The gradient concentration of each component was taken as the abscissa, and the average value of the three measured peak areas was taken as the ordinate, in order to obtain the quantitative linear relationship of each component. The content of 2AP was calculated according to the linear equation using the peak area, and then other volatile substances were semi-quantitated according to the content of 2AP [26]. The calculation formula was Xi = (Ai/As) × Cs (Xi was the content of the substance to be tested; Ai was the peak area of the substance to be tested; As was the peak area of 2AP in standard solution; Cs was the content of 2AP in the sample).

#### 2.2.3. Determination of Soil Samples

##### Sample Collection

Soil samples were collected in October 2021. Three initial soil samples (0–20 cm) were randomly collected from each plot by a 5 cm diameter soil auger, and then mixed as one soil sample for each plot. Plant roots and other visible foreign bodies were removed from soil samples via sieving (<2 mm, <0.20 mm). The soil samples were brought back to analyze soil physico-chemical properties after air drying.

##### Sample Analysis

Soil pH was measured using a pH/conductivity meter (FE28, China; the soil:water ratio was 1:2.5). After weighing the fresh weight, the soil samples were oven-dried at 105 °C for 24 h, and weighed again to calculate the soil moisture (SM). Soil organic matter (SOM) was determined with a total organic carbon analyzer (Multi N/C 3100, Jena, Germany), and bulk density (SBD, g cm^−3^) was measured as the ratio of soil dry weight and soil volume. The soil alkali-hydrolyzed nitrogen (SAN) was determined using the alkaline hydrolysis diffusion method. Soil available phosphorus (soil Olsen-P, SOP) was assessed using Bray’s method (UV2310 II, Shanghai, China). Soil available potassium (SAK) was determined using flame photometry (6400A, Changsha, China).

#### 2.2.4. Determination of Ecosystem Environment

The soil and air temperature (ST and AT, respectively) were measured with a thermocouple probe that connected to the portable soil carbon dioxide flux measurement system (Li-8100, Li-Cor, Inc., Lincoln, NE, USA). The AT was measured at the canopy of *Pandanus amaryllifolius* (50 cm), and the ST was measured at 20 cm, where *Pandanus amaryllifolius* roots were widely distributed at this depth. The ST and AT were measured in December 2020, March 2021, June 2021 and September 2021 (once every season). The annual average of AT and ST were used to represent the ecosystem micro-environment indicators for the experimental site.

#### 2.2.5. Statistical Analysis

The *t*-test was used to determine the differences in experimental indicator and volatile substances (i.e., soil physico-chemical properties, micro-environment indicators, the component numbers and relative contents of volatile categories) between the intercropping and monoculture patterns. Partial least squares discriminant analysis (PLSDA) was used to distinguish the overall difference in volatile substances’ relative content between the monoculture and intercropping pattern. The network interaction analysis of soil properties and volatile substances were used to measure the correlation between component numbers or relative contents of volatile substance in different categories and soil properties, as well as the correlation between component numbers or relative contents of volatile substance in different categories. Redundancy analysis (RDA) of environmental factors and relative contents of volatile substance categories was conducted, which was assessed from 999 iterations based on Monte Carlo permutations. Data analyses were performed using SPSS 23.0, SAS V8 and Canaco 5.0. The graphs were plotted using Origin 2021b and Cytoscape V3.8.2.

## 3. Results and Discussion

### 3.1. Effects of Intercropping Pattern on Soil Physico-Chemical Properties and Ecosystem Micro-Environment

The *t*-test was used to explicate differences in soil physico-chemical properties among different cultivation patterns. Compared with *Pandanus amaryllifolius* monoculture, the soil pH in the intercropping pattern was significantly decreased by 1.00 (*p* < 0.05), while SBD, SAN and SOP were significantly increased by 0.24 g cm^−3^, 31.01 mg kg^−1^ and 10.49 mg kg^−1^, respectively (*p* < 0.05). There was no difference between the intercropping and *Pandanus amaryllifolius* monoculture patterns on SM, SOM and SAK (Table 1). However, compared with *Pandanus amaryllifolius* monoculture, AT tended to increase by 1.02 °C under the intercropping pattern (*p* < 0.1).

Soil is the basis for crop growth. Improvements in soil micro-environment indices and physico-chemical properties have positive effects on promoting crop physiological activity and increasing productivity [27]. There is a significant impact on the soil properties of tropical perennial plantations under changes in cultivation patterns, especially the intercropping pattern. Previous studies showed that the intercropping pattern could improve the contents of SAP, SAN, SAK and SOM, and promote benign interactions between crops [28,29]. Compared with the *Pandanus amaryllifolius* monoculture, the intercropping significantly increased the contents of SAN and SOP in this study (Table 1), which is consistent with the results of previous studies [28,29]. Furthermore, the intercropping pattern in the current study significantly improved SBD, which may be related to the root characteristics and activities of the two crops [30]. An increase in SBD usually implies a decrease in soil organic matter content, but there was no significant difference between the SOM of the intercropped and monoculture cultivations in this study (Table 1).

### 3.2. Effects of Intercropping Pattern on Pandanus amaryllifolius Volatile Substances

#### 3.2.1. Comparison of *Pandanus amaryllifolius* Volatile Substances under Different Cultivation Patterns

The component numbers of *Pandanus amaryllifolius* volatile substances in different categories under monoculture and intercropping patterns were measured via GC–MS. There were 68 main volatile components in *Pandanus amaryllifolius* leaves, which were divided into 10 categories, including alcohols, pyrroles, ketones, esters, aldehydes and ketones, furans, furanones, acids, hydrocarbons and phenols (Table A1). The results of *t*-tests showed that compared with *Pandanus amaryllifolius* monoculture, the component number of esters significantly increased by 6.20%, while the component number of ketones significantly decreased by 4.26% (*p* < 0.05, Figure 1a). Compared with the *Pandanus amaryllifolius* monoculture, the relative contents of pyrroles, esters and furanones significantly increased by 8.83%, 2.30% and 8.27%, respectively (*p* < 0.05), while the relative contents of ketones, furans and hydrocarbons significantly decreased by 1.01%, 10.55% and 9.16% (*p* < 0.05), respectively, under the intercropping pattern. There were no differences found between monoculture and intercropping in the relative contents of alcohols, acids, aldehydes, ketones and phenols in *Pandanus amaryllifolius* leaves (Figure 1b).

The Q^2^ of PLSDA analysis on the relative contents of volatile substances in different categories under monoculture and intercropping pattern was 0.95, indicating that the model prediction results of this experiment complied with the requirements. There was a significant difference in the relative contents of volatile substances under the *Pandanus amaryllifolius* monoculture and intercropping patterns in this study (Figure 2), indicating that the different cultivation patterns significantly affected the synthesis and accumulation of volatile substances in *Pandanus amaryllifolius* leaves.

#### 3.2.2. Co-Occurrence Network Analysis between Soil Properties and Volatile Substances of *Pandanus amaryllifolius*

Co-occurrence network analysis was used to determine the co-occurrence patterns of soil properties, component numbers and relative contents of *Pandanus amaryllifolius* volatile substances in different categories, based on strong and significant correlations (Figure 3).

Overall, different cultivation patterns showed a remarkable effect on the association networks of soil properties and component numbers of volatile substances in different categories. The values of average path length (APL), average connectivity (*avgK*), average clustering coefficient (*avgCC*) and graph density in these empirical networks of component numbers of volatile substances in different categories were 1.86, 0.58, 2.46 and 0.21, respectively (Table 2). More negative co-occurrence relationships were shown between the component numbers of volatile substances in different categories, while more positive co-occurrence relationships were shown from soil properties to the component numbers of volatile substances in different categories in the network graph (Figure 3a). The values of average path length (APL), average connectivity (*avgK*), average clustering coefficient (*avgCC*) and graph density in these empirical networks of relative contents of *Pandanus amaryllifolius* volatile substances in different categories were 2.11, 0.71, 5.20 and 0.37, respectively (Table 2). More negative co-occurrence relationships were shown between relative contents of *Pandanus amaryllifolius* volatile substances in different categories, while more positive co-occurrence relationships were shown from soil properties to relative contents of *Pandanus amaryllifolius* volatile substances in different categories, which indicated that the improvements in soil properties were conducive to increasing component numbers and relative contents of volatile substances in different categories.

#### 3.2.3. Correlation among Soil Physico-Chemical Properties, Ecosystem Micro-Environment and Volatile Substances of *Pandanus amaryllifolius*

RDA analysis was used to ensure the relationships between micro-environment indicators and volatile substances. The results showed that the component numbers and relative contents of volatile substances in different categories responded differently to changes in environmental factors under different cultivation patterns (Figure 4).

The soil pH, among the soil properties, was consistently related to changes in the component numbers of esters, ketones and furanones (Figure 4a and Table 3). Soil pH, SOP and AT were associated with changes in the relative contents of pyrroles, esters, furanones, ketones, furans and hydrocarbons (Figure 4b and Table 3). For the pyrroles, hydrocarbons and alcohols, which affect the key aroma components of the quality of *Pandanus amaryllifolius*, there was a significant positive correlation between AT, SOP and the relative contents of pyrroles, hydrocarbons and alcohols, while there was an opposite effect on the correlations between pH and relative content of pyrroles, hydrocarbons and alcohols.

### 3.3. Key Regulated Factors of Intercropping Pattern on Volatile Substances

#### 3.3.1. Key Regulated Factors of Intercropping Pattern on the Component Numbers of *Pandanus amaryllifolius* Volatile Substances in Different Categories

*Pandanus amaryllifolius* is a perennial herb spice crop, which is mainly used for its leaves. The main volatile compounds affecting the quality of *Pandanus amaryllifolius* are alcohols, pyrroles and hydrocarbons, according to the analysis of its leaf compounds from a previous study [31]. In particular, 2AP is the most important substance in pyrroles, which is also the most important volatile aroma substance of *Pandanus amaryllifolius*, Thai fragrant rice and other crops [32]. It is generally believed that soil properties and planting patterns are the two main factors that affect crop quality [33]. On the one hand, the lack of some soil elements is a key factor limiting the synthesis of certain substances [34], while some excessive soil elements also have a negative feedback effect on the synthesis of certain substances [35]. Improvements in soil nutrients could promote crop productivity and quality by increasing the synthetic substrates of crops [36]. On the other hand, the increased crop species richness could promote the growth and quality of crops by improving soil biological activity, and accelerating the adaptability of crops to the environment under the intercropping pattern [21,37]. Furthermore, there is a competitive relationship between various crops on soil nutrients, which can change the physiological metabolism process of crops, and have a greater impact on crop quality by increasing parts of compound synthesis paths under the intercropping pattern [38]. Therefore, the numbers of compound types in ketones and furanones were significantly and negatively correlated with SAN, SOP and SBD, while there was a positive correlation between esters and the above soil property indexes under the intercropping pattern (Figure 3b) in the current study; these findings indicate that the increased contents of SBD, SAN and SOP under the intercropping pattern may be the main reason for the decrease in ketone and furanone types, and the increase in ester types.

#### 3.3.2. Key Regulated Factors of Intercropping Pattern on the Relative Contents of *Pandanus amaryllifolius* Volatile Substances in Different Categories

The volatile substances of spice crops are not only determined by gene structure, but also are affected by the growth environment and cultivation methods. Firstly, previous studies proved that the aromatic substance of spearmint increased when the climate and altitude were both appropriate [39]. Compared with the *Pandanus amaryllifolius* monoculture, air temperatures under the *Hevea brasiliensis* and *Pandanus amaryllifolius* intercropping ecosystem showed an upward trend in the current study (Table 1). There was a positive correlation between pyrrole substances’ relative contents in *Pandanus amaryllifolius* leaves and air temperature (Figure 3b), indicating that the increase in air temperature after intercropping may be one of the main factors to improve the physiological activity of the leaves of *Pandanus amaryllifolius*, and accelerate the synthesis rate of pyrrole substances. Secondly, research shows that *Rosa rugosa* and *Elymus dahurica* strengthen the absorption of soil nutrients by increasing the number of root tillers after intercropping with other crops, which has played a potential role in promoting their aromatic components content and productivity [36,40]. The increased contents of SAN and SOP, as well as the positive correlation between SAK, SOP and pyrrole content (Figure 3b and Figure 4b), indicate that the increase in soil nutrients may be another main reason for the accelerated accumulation of pyrrole substances in the current study. Thirdly, the positive correlation between the hydrocarbon content and pH in this study shows that an acidic soil environment after intercropping may be unfavorable for the synthesis and accumulation of hydrocarbons in *Pandanus amaryllifolius* leaves. Moreover, the negative correlation between hydrocarbon content and SAN and SOP content indicates that an increase in soil nutrients under an intercropping pattern may have a significant negative feedback effect on hydrocarbon synthesis. Overall, the increase in soil nutrient content and the change in ecosystem micro-environment under an intercropping pattern are potential factors affecting the content and quality of volatile substances in *Pandanus amaryllifolius* leaves. The results of this study provide a theoretical basis for the application and promotion of the *Pandanus amaryllifolius* and *Hevea brasiliensis* incropping productive model, by clarifying the impact of environmental factors on the main qualities of *Pandanus amaryllifolius* under an intercropping pattern.

## 4. Conclusions

The volatile substances of *Pandanus amaryllifolius* leaves significantly changed after intercropping with *Hevea brasiliensis*. The intercropping pattern decreased component numbers of ketones and furanones, while it had the opposite effect on esters. The relative contents of ketones, furans and hydrocarbons decreased, but the contents of pyrroles, esters and furanones significantly increased under the intercropping pattern. The increased contents of SBD, SOP and SAN, and the decreased pH under the intercropping pattern, may be the main reasons for significantly increased pyrrole and decreased hydrocarbon relative contents. Intercropping with *Hevea brasiliensis* significantly promoted the synthesis and accumulation of the main volatile substance categories of *Pandanus amaryllifolius* leaves. The results of this study are conducive to the establishment of high-quality cultivation patterns for *Pandanus amaryllifolius*.

## Figures and Tables

**Figure 1 foods-12-00888-f001:**
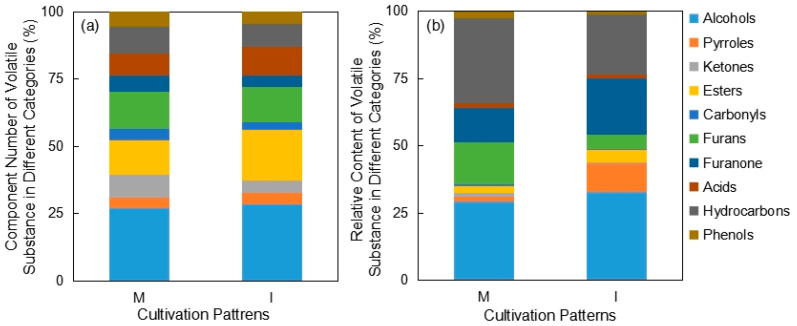
Effects of monoculture and intercropping patterns on the component numbers (**a**) and relative contents (**b**) of volatile substance in different categories. *n* = 10. M: *Pandanus amaryllifolius* monoculture, I: *Pandanus amaryllifolius* and *Hevea brasiliensis* intercropping.

**Figure 2 foods-12-00888-f002:**
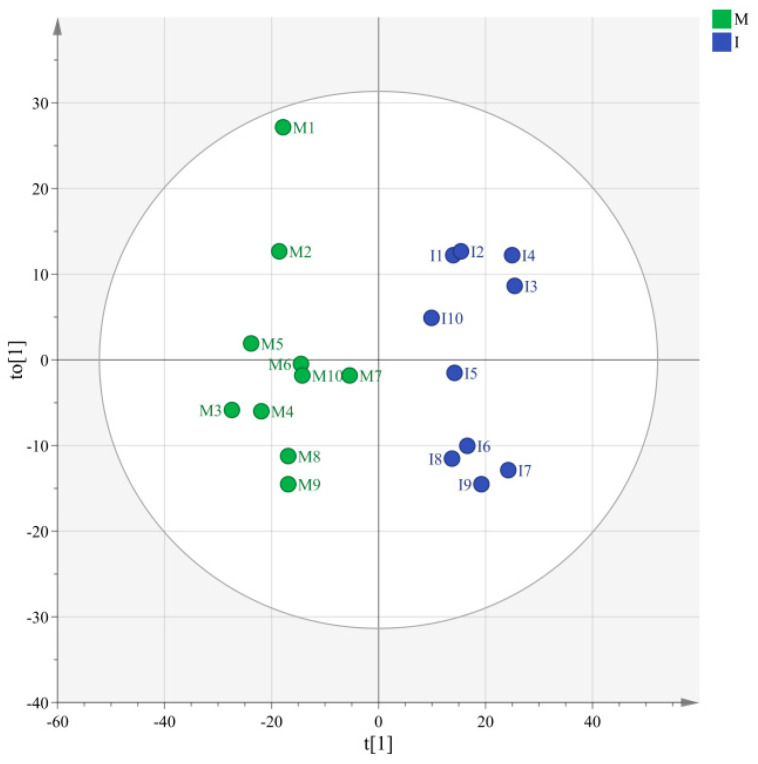
Effects of monoculture and intercropping patterns on the component numbers and relative contents of volatile substances in different categories (PLSDA). See Figure 1 for treatment abbreviations.

**Figure 3 foods-12-00888-f003:**
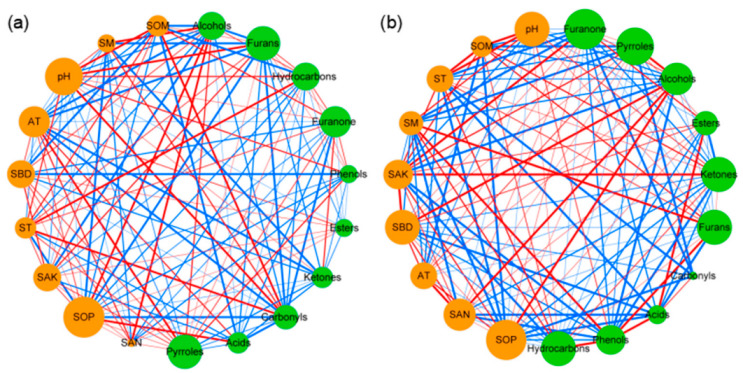
Network interaction diagram of the component numbers and relative contents of volatile substances in different categories in monoculture (**a**) and intercropping (**b**). Red lines indicate positive correlations, while blue lines indicate negative correlations. The thickness of the lines represents the correlation size. The size of the points represents the magnitude of relative types and contents of volatile components.

**Figure 4 foods-12-00888-f004:**
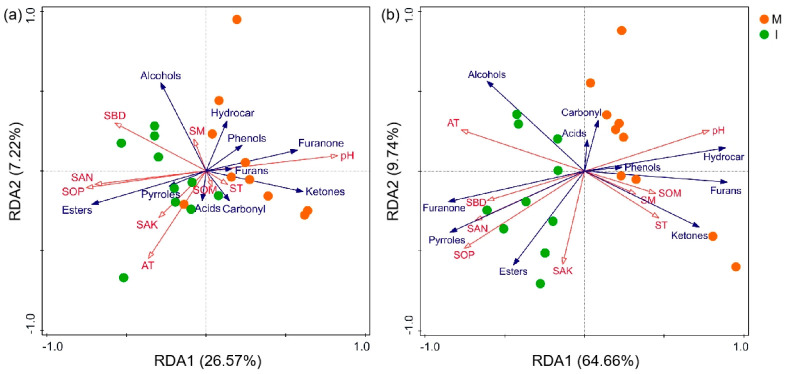
The redundancy analysis (RDA) of the component numbers (**a**) and relative contents (**b**) of volatile substances in different categories and environmental factors under different cultivation patterns. Blue arrows indicate volatile substances, red arrows indicate soil properties. See Figure 1 for treatment abbreviations.

**Table 1 foods-12-00888-t001:** Soil physico-chemical properties under monoculture and intercropping patterns.

Soil Physico-Chemical Properties	Cultivation Patterns	
	*Pandanus amaryllifolius* Monoculture	*Pandanus amaryllifolius* and *Hevea brasiliensis* Intercropping
AT (°C)	33.50 ± 0.32 b	34.52 ± 0.31 a
ST (°C)	20.24 ± 0.10 a	20.10 ± 0.09 a
pH	7.15 ± 0.08 a	6.15 ± 0.03 b
SM (%)	18.88 ± 0.55 a	17.62 ± 0.79 a
SBD (g cm^−3^)	1.78 ± 0.06 b	2.02 ± 0.08 a
SOM (g kg^−1^)	17.44 ± 2.08 a	15.25 ± 1.17 a
SAK (mg kg^−1^)	38.13 ± 1.60 a	41.85 ± 1.45 a
SAN (mg kg^−1^)	51.95 ± 2.65 b	82.96 ± 2.02 a
SOP (mg kg^−1^)	5.44 ± 0.24 b	15.93 ± 0.67 a

M: *Pandanus amaryllifolius* monoculture; I: *Pandanus amaryllifolius* and *Hevea brasiliensis* intercropping. Mean ± SE. Different letters represent significant differences at *p* < 0.05; AT: air temperature, ST: soil temperature; SM: soil moisture; SBD: soil bulk density; SOM: soil organic matter; SAK: soil available potassium; SOP: soil available phosphorus; SAN: soil alkali-hydrolyzed nitrogen.

**Table 2 foods-12-00888-t002:** Topological properties of co-occurring soil and volatile substances networks obtained in different cultivation patterns.

Network Metrics	Component Numbers of Volatile Substances in Different Categories	Relative Contents of Volatile Substances in Different Categories
Number of Nodes	19	19
Number of Edges	130	135
Number of Positive Correlations	60	61
Number of Negative Correlations	70	74
Percentage of the Positive Link (P%)	46.15	45.19
P% from Soil Properties to Volatile Substances Categories	52.59	53.90
Average Connectivity (*avgK*)	2.46	5.20
Average Clustering Coefficient (*avgCC*)	0.58	0.71
Average Path Length (APL)	1.86	2.11
Graph Density	0.21	0.37

**Table 3 foods-12-00888-t003:** The results of redundancy analysis (RDA) on the component numbers and relative contents of volatile substances in different categories and soil properties under different cultivation patterns.

Sequence	Component Numbers of Volatile Substances in Different Categories			Relative Contents of Volatile Substance in Different Categories		
	Explains	*F* Value	*p* Value	Explains	*F* Value	*p* Value
pH	13.50	2.80	0.04	40.90	12.40	0.01
AT	4.40	0.90	0.45	18.30	7.60	0.01
SOP	3.40	0.60	0.60	5.90	2.70	0.04
SOM	1.80	0.30	0.88	3.30	1.60	0.18
SBD	4.20	0.90	0.52	2.10	1.00	0.39
SAK	4.90	1.00	0.42	2.00	0.90	0.42
SAN	2.60	0.50	0.73	3.00	1.50	0.19
ST	1.20	0.20	0.95	2.10	1.00	0.38
SM	3.90	0.80	0.50	1.90	0.90	0.48

## Data Availability

Data used for this analysis are available at: https://datadryad.org/stash/share/2K8HTtaRlFynVoaXZkT1_68ZMb2nLZ5c-TaUas0rTzs.

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
