# Peer review of "Effects of *Hevea brasiliensis* Intercropping on the Volatiles of *Pandanus amaryllifolius* Leaves"

_foods, 2023, doi:10.3390/foods12040888_

Round 1

Reviewer 1 Report

The study by Lu et al. deals with the impact of intercopping between Hevea brasiliensis and Pandanus amaryllifolius on the diversity and relative contents of volatiles in P. amaryllifolius leaves.

The objective of the work is interesting, as are the results obtained. However, I note serious problems in the quality of the manuscript:

1) the Journal format was not followed, neither in the headlines nor in the quotes and references.

2) the English language is often wrong. It is clear that the manuscript has not been checked by a native English speaker.

3) numerous inaccuracies and typos are scattered throughout the text.

Some abbreviations are used in the abstract which are not explained (SBD, SAN, SOP).

Among the keywords, intercopping is not mentioned, as well as the names of the species investigated.

In the Introduction section, chlorophyll and chlorophyllin are presented as volatile compounds. This requires an explanation.

The rules of the binomial nomenclature are often not respected.

Some of the problems I found in the manuscript are indicated in the attached pdf file.

I suggest the authors to review the manuscript carefully before submitting it again to the Journal. The advice of a native English speaker is required.

Author Response

Dear Reviewer:

Thanks for your careful review and valuable comments on this manuscript. Authors have carefully revised and supplemented according to your comments, and all the revised parts are marked with trace in the new manuscript. For the attached modification opinions, we modify and supplement them item by item as follows: 

  1. The Journal format was not followed, neither in the headlines nor in the quotes and references.

Thanks for your suggestion, the author has revised the format of the manuscript, and then  checked the format of the manuscript according to the requirements of the Foods.

  1. The English language is often wrong. It is clear that the manuscript has not been checked by a native English speaker.

Thanks for your suggestion, the English version of the manuscript has been revised in detail by the author to make it easier for readers.

  1. Numerous inaccuracies and typos are scattered throughout the text.

Thanks for your suggestion, the author has revised the text of the manuscript in detail, and the manuscript has been reviewed by experts of the same profession to ensure that the manuscript will not have inaccuracies and typos errors

  1. Some abbreviations are used in the abstract which are not explained (SBD, SAN, SOP).

Thanks for your suggestion, the relevant abbreviations has been modified in the new manuscript, please see line 23-24.

  1. Among the keywords, intercopping is not mentioned, as well as the names of the species investigated.

Thanks for your suggestion, the intercopping pattern has been added to the keyword, and and the names of the species has been added to the title. Please see line 1 and 38-39.

  1. In the Introduction section, chlorophyll and chlorophyllin are presented as volatile compounds. This requires an explanation.

Thank you for your reminder, the author inputs the name of “Phytol” into “Chlorophyll” after checks the original data. The revision has been completed in the new manuscript and make sure there are no mistakes, please see line 49, 55, 89.

  1. The rules of the binomial nomenclature are often not respected.

Thanks for your suggestion, the author has revised the non-standard application of binomial nomenclature in detail. Please see line 42 and 64.

  1. Some of the problems I found in the manuscript are indicated in the attached pdf file.

Thanks for your detailed comments. The author has carefully checked all the text and data information in the new manuscript. 

  1. I suggest the authors to review the manuscript carefully before submitting it again to the Journal. The advice of a native English speaker is required.

Thanks for your valuable comments. The author invited relevant experts to revise the text, hoping to make it easy for readers to read.

Reviewer 2 Report

This manuscript describes the Hevea brasiliensis and Pandanus amaryllifolius intercropping effect on volatile substances in the Pandanus amaryllifolius leaves and the key regulatory factors of volatile substances. The conclusions of this experiment indicated that the Hb intercropping with Pa could not only improve the soil properties, but also significantly increase the relative content of the main volatile substances in Pa leaves, which could provide a theoretical basis for the application and promotion of high-quality production pattern of Pa. 

The manuscript is generally well prepared (even if it requires many corrections), but there are some doubts about the research that is due to the working methodology and the very short study time to state such conclusions.

Thus, “The experimental plot was constructed in October 2020 (see Line L 104), and healthy leaves samples were selected and collected in October 2021 (see Line L 113)”. 

The authors should provide more credible arguments to support the conclusions presented in their study. Even if statistical data are presented, the possibility of improving the soil properties, and increasing the relative content of the main volatile substances in Pa leaves in such a short time, due to Hb+Pa intercropping (versus to M), is debatable.

For this purpose, the work methodology should be better clarified, the experimental conditions, the size of the plants (for example Hb, what size did it reach after one year, and how Hb and Pa influenced the physicochemical properties of the soil and volatile components, where and how AT, ST, etc. determinations were performed etc.). 

Ambiguities should be avoided.  

For example, see Line L 106-110:

“... each planting pattern: Hb monoculture (A), Pa monocropping (M), Hb and Pa intercropping (I)...”, and then “... (only M and I planting pattern were used in this study)”.

If you did not include ‘Hb monoculture (A)’ in the study, remove this redundant ‘planting pattern’ from the manuscript, you only create confusion. 

L 156-159: “The soil and air temperature (ST and AT, respectively) were measured by the thermocouple probe which connected to the portable soil carbon dioxide flux measurement system (Li-8100, Li-Cor, Inc., Lincoln, NE, USA)”. Details required! Avoiding subjective aspects. Revise and improve the uniformity of determinations and the observance of a methodology that confers the correctness of the data (i.e., how, when, how much, where - in which place/position the measurements were carried out) 

Please also check the Lines L 160-163:

“2.5. Data analysis

One-way ANOVA was used to determine the differences of experimental indicator (i.e. soil physicochemical properties, types and relative contents of volatile components) between intercropping and monocropping patterns. Partial least squares discriminant...” 

Here you are referring to two treatments, respectively intercropping and monocropping. In the rest of the manuscript, instead of ‘monocropping’, use ‘monoculture’. Avoid the confusion you can cause to the readers and be consistent in the use of terms. Use monoculture everywhere for Pa. 

For the comparison of two means (i.e., M and I) in statistics, the t-test is recommended. However, you can leave the one-way ANOVA for the results in Table 1 because the results are the same between the t-test and one-way ANOVA.

But check once again the correctness and rigor of the data and the table-text correspondence. 

For example, for AT (℃) your data in Table 1 are 33.50±0.32b (M) 34.52±0.31a (I), and in the text, you say that there is a difference of 0.57 (see L17-180: „However, compared with Pa monoculture, AT tended to increased by 0.57℃ under intercropping pattern (P<0.1)”. 

Review the tables and figures 

For example, in Table 1 - at the top, in the head of the table, write ‘Monoculture’ instead of ‘M’, and ‘Intercropping’ instead of ‘I’ (there is enough space, and thus you facilitate an easier understanding of the data and manuscript by the readers). 

Under Table 1, you use the syntax “Error bars indicate SE”. But you do not have error bars (they can only appear in graphics)!!! You are probably referring to the standard error - SE (or the standard error of the mean - SEM? Please verify).

Anyway, please correct: your values presented and you refer to in Table 1 are mean ± SE 

L 206-207 Figure 2. “Effects of monoculture and intercropping patterns on types and relative contents of volatile substances (PLSDA). See Fig. 2 for treatment abbreviations” (!!!). 

There is the same mistake in Figure 4 (“See Fig. 2 for treatment abbreviations” - L239)!!! 

L 292-297: “Compared with Pa monoculture, air temperature under the Hb anf Pa intercropping ecosystemThere was shown an upward trend in current study (Table 1) and a positive correlation between the pyrrole substances relative content in Pa leaves and air temperature (Figure 3b), indicates that the increase of air temperature after intercropping may be one of the main factors to improve the physiological activity of the leaves of Pa and accelerate the synthesis rate of pyrrole substances”.

There is only one incoherent sentence (it needs to be revised and separated into clear sentences. In addition, there are several typographical errors ('anf' instead of 'and'), and lack of spaces or punctuation marks (i.e., “ecosystemThere”). 

L 298: Use italics for species names, i.e., for Rosa rugosa and Elymus dahurica (see the text “Second, the research shows that Rosa rugosa and Elymus dahurica strengthen...”). 

Review the whole content of the manuscript, the correctness of the text and notions, and avoid typos, grammar, and spelling errors. I.e., see L 189-190: “There were 10 types of volatile components were detected, including alcohols, ...”.

Consequently, the correctness of the text, data, notions, spaces between the values of some parameters and their units of measure, etc. must be checked in the entire manuscript.

Author Response

Dear Reviewer:

Thanks for your careful review and valuable comments on this manuscript. Authors have carefully revised and supplemented according to your comments, and all the revised parts are marked with trace in the new manuscript. For the attached modification opinions, we modify and supplement them item by item as follows: 

  1. “The experimental plot was constructed in October 2020 (see Line L 104), and healthy leaves samples were selected and collected in October 2021 (see Line L 113)”.

Sorry for the unclear description confused the reviewer. Pandan was generally harvested 10 months after seedlings are transplanted to the field. Pandan can be harvested all year round, and the frequency of harvesting is usually once every month in summer and once every two months in winter. Thus, the author chose to take samples one year after the seedlings were transplanted to the field (the time of transplanting Pandan seedling is October 2020). Please see line 103-112.

  1. The authors should provide more credible arguments to support the conclusions presented in their study. Even if statistical data are presented, the possibility of improving the soil properties, and increasing the relative content of the main volatile substances in Pa leaves in such a short time, due to Hb+Pa intercropping (versus to M), is debatable.

Thanks for your comments. There was a significant difference in category of volatile substances in Pa leaves between intercropping and monocropping. The author believes that there are three main factors in affecting the relative content of volatile substances under intercropping pattern: 1. Intercropping affecting the nutrient demand, enzyme activity and synthesis rate of volatile substances in the Pa leaves by changing soil properties (nutrients contentin) and environmental factors (radiation and temperature) ; 2. Intercropping affects the synthesis of volatile substances in Pa leaves by changing the type and activity of the exudates in the rhizosphere of crops, forming the interaction relationship between crops such as enhanced competition or synergetic absorption; 3. Intercropping affects the synthesis of volatile substances by reshaping the structure and function of soil microbiota and the interaction between crop root, soil and microorganisms. It is still unclear which of the three possible regulatory pathways has the greatest impact, thus a rigorous field experiment will be  established to further explore the regulation mechanism of volatile substances in Pa leaves.

  1. For this purpose, the work methodology should be better clarified, the experimental conditions, the size of the plants (for example Hb, what size did it reach after one year, and how Hb and Pa influenced the physicochemical properties of the soil and volatile components, where and how AT, ST, etc. determinations were performed etc.).

Thanks for your comments, the author describes in detail the experimental process, as well as the determination method and timing of indicators in the new manuscript. Please see line 130-189.

  1. Ambiguities should be avoided.For example, see Line L 106-110:“... each planting pattern: Hb monoculture (A), Pa monocropping (M), Hb and Pa intercropping (I)...”, and then “... (only M and I planting pattern were used in this study)”. If you did not include ‘Hb monoculture (A)’ in the study, remove this redundant ‘planting pattern’ from the manuscript, you only create confusion.

Thanks for your comments. The author has deleted some ambiguous descriptions. Please see line 121-127.

  1. L 156-159: “The soil and air temperature (ST and AT, respectively) were measured by the thermocouple probe which connected to the portable soil carbon dioxide flux measurement system (Li-8100, Li-Cor, Inc., Lincoln, NE, USA)”. Details required! Avoiding subjective aspects. Revise and improve the uniformity of determinations and the observance of a methodology that confers the correctness of the data (i.e., how, when, how much, where - in which place/position the measurements were carried out)

Thanks for your comments. The author has added relevant information in the new manuscript. Please see line 181-189.

  1. Please also check the Lines L 160-163:“2.5. Data analysisOne-way ANOVA was used to determine the differences of experimental indicator (i.e. soil physicochemical properties, types and relative contents of volatile components) between intercropping and monocropping patterns. Partial least squares discriminant...” For the comparison of two means (i.e., M and I) in statistics, the t-test is recommended. However, you can leave the one-way ANOVA for the results in Table 1 because the results are the same between the t-test and one-way ANOVA.

Thanks for your comments, the author has revised the statistical method according to the  suggestions. Please see line 191-203.

  1. Here you are referring to two treatments, respectively intercropping and monocropping. In the rest of the manuscript, instead of ‘monocropping’, use ‘monoculture’. Avoid the confusion you can cause to the readers and be consistent in the use of terms. Use monoculture everywhere for Pa.

Thanks for your comments. The author has unified the full text description of singal planting pattern as “monoculture ” according to the suggestions in the new manuscript. Please see line 121-127. 

  1. But check once again the correctness and rigor of the data and the table-text correspondence.For example, for AT (℃) your data in Table 1 are 33.50±0.32b (M) 34.52±0.31a (I), and in the text, you say that there is a difference of 0.57 (see L17-180: „However, compared with Pa monoculture, AT tended to increased by 0.57℃ under intercropping pattern (P<0.1)”.

Sorry for the author's carelessness. The author has checked the text and data in detail to ensure that there are no data and description errors in the new manuscript. Please see line 213.

  1. Review the tables and figures. For example, in Table 1 - at the top, in the head of the table, write ‘Monoculture’ instead of ‘M’, and ‘Intercropping’ instead of ‘I’ (there is enough space, and thus you facilitate an easier understanding of the data and manuscript by the readers).Under Table 1, you use the syntax “Error bars indicate SE”. But you do not have error bars (they can only appear in graphics)!!! You are probably referring to the standard error - SE (or the standard error of the mean - SEM? Please verify).Anyway, please correct: your values presented and you refer to in Table 1 are mean ± SE.

Thanks for your comments. Author has revised the title and notes of Table 1 according to the comments. Please see line 214-219.

  1. L 206-207 Figure 2. “Effects of monoculture and intercropping patterns on types and relative contents of volatile substances (PLSDA). See Fig. 2 for treatment abbreviations” (!!!).

Thanks for your comments. Author has revised the notes of Figure 2 according to the comments. Please see line 256-257.

  1. There is the same mistake in Figure 4 (“See Fig. 2 for treatment abbreviations” - L239)!!!

Thanks for your comments. Author has revised the notes of Figure 4 according to the comments. Please see line 294-297.

  1. L 292-297: “Compared with Pa monoculture, air temperature under the Hb anf Pa intercropping ecosystemThere was shown an upward trend in current study (Table 1) and a positive correlation between the pyrrole substances relative content in Pa leaves and air temperature (Figure 3b), indicates that the increase of air temperature after intercropping may be one of the main factors to improve the physiological activity of the leaves of Pa and accelerate the synthesis rate of pyrrole substances”.There is only one incoherent sentence (it needs to be revised and separated into clear sentences. In addition, there are several typographical errors ('anf' instead of 'and'), and lack of spaces or punctuation marks (i.e., “ecosystemThere”).

Thanks for your comments. Author has revised the rext according to the comments. Please see line 339-340.

  1. L 298: Use italics for species names, i.e., for Rosa rugosa and Elymus dahurica (see the text “Second, the research shows that Rosa rugosa and Elymus dahurica strengthen...”).

Thanks for your comments. Author has revised the rext according to the comments. Please see line 345-346.

  1. Review the whole content of the manuscript, the correctness of the text and notions, and avoid typos, grammar, and spelling errors. I.e., see L 189-190: “There were 10 types of volatile components were detected, including alcohols, ...”.

Thanks for your comments. Author has revised the rext according to the comments, and then revised and checked similar descriptions in the full text. Please see line 236-237.

  1. Consequently, the correctness of the text, data, notions, spaces between the values of some parameters and their units of measure, etc. must be checked in the entire manuscript.

Thanks for your comments. The author has carefully checked all the text and data information in the new manuscript.

Reviewer 3 Report

The authors of the manuscript presented interesting results of their research work , which may have application potential. My comments concern the following questions:

1. Why was ethanol chosen for the extraction of volatile compounds not another method like headspace technique.

2. Why were not given the individual components of the volatile fractions in the manuscript but only groups of compounds, especially since the authors compared the results obtained with the database and wrote in line 132 "each volatile component" , and not "types of volatile components" as in line 189? Therefore, in my opinion, the phrase of volatile components should be avoided in the text, which can be confusing, but rather groups of volatile components.

3. Was the extraction temperature at 50°C not too high for volatile compounds?

Author Response

Dear Reviewer:

Thanks for your careful review and valuable comments on this manuscript. Authors have carefully revised and supplemented according to your comments, and all the revised parts are marked with trace in the new manuscript. For the attached modification opinions, we modify and supplement them item by item as follows: 

  1. Why was ethanol chosen for the extraction of volatile compounds not another method like headspace technique.

Thanks for your comments. The author's research team had already try to use headspace technique during the construction of analytical methods for volatile substances, but the repeatability of this method does not meet the requirements and it is disadvantageous to the requirements of experiment on control error. Therefore, the stable direct injection method was selected to be used in the research work in the early stage of this experiment. This study is a preliminary experiment to explore the changes of volatile components of pandan under different cultivation poatterns. Only the difference of volatile substance categories can be judged up to present. In future research, we will try to optimize the headspace injection method to eliminate the interference of solvent on volatile components.

  1. Why were not given the individual components of the volatile fractions in the manuscript but only groups of compounds, especially since the authors compared the results obtained with the database and wrote in line 132 "each volatile component" , and not "types of volatile components" as in line 189? Therefore, in my opinion, the phrase of volatile components should be avoided in the text, which can be confusing, but rather groups of volatile components.

Thanks for your valuable suggestions. Shall we use “volatile components category” instead of “volatile components” in the full text to reduce confusion? If this is not rigorous enough, the author will use “groups of volatile components” instead of “volatile components” as suggested by the reviewer.

  1. Was the extraction temperature at 50°C not too high for volatile compounds?

Thanks for your comments. The author's research team had already tested the method of extracting volatile substances from pandan at different temperatures (30 oC, 40oC, 50oC, and 60oC) in the previous experiment. The results showed that the type and content of volatile substances extracted at 30 oC were the least. Although the type and content of volatile substances extracted at 40 oC were significantly higher than 30 oC, the key substance (2-acetyl-1-pyrroline) of pandan was not effectively extracted. The extraction at 60 oC would lead to the rapid blackening of pandan leaves (possibly due to the decomposition or destruction of some substances at high temperature). Thus, 50 oC was adopted as the  extraction temperature in the preliminary work of this study.

Round 2

Reviewer 1 Report

The authors have carefully revised the original manuscript, both in terms of content and form.

Unfortunately some small errors are still present in the manuscript, which must be recorded before publication. Some of them are highlighted in blue/green in the attached pdf file.

Author Response

Thanks for your suggestion, the author has revised the format of the manuscript, and then  checked the format of the manuscript according to the requirements of the Foods. Please refer to the new manuscript for details.

Reviewer 2 Report

Dear Authors,

We noted that you approached the reported aspects carefully. As a result of your work, the manuscript is much more consistent and suitable for readers.

Some grammar, spelling mistakes, or typos (punctuation marks, capital letters for common nouns, etc.), which appear from the start of the new version of the manuscript (in abstract, keywords - see "Intersociation', instead of 'Interassociation', etc.) can be solved in the editing stages and processes.

Therefore, congratulations on your work, and continued success!

Author Response

Thanks for your suggestion, the author has revised the format of the manuscript, and then  checked the format of the manuscript according to the requirements of the Foods. Please refer to the new manuscript for details